# Relationship between Carotid Intima-Media Thickness, Periodontal Disease, and Systemic Inflammation Biomarkers in an Adult Population

**DOI:** 10.3390/biomedicines12071425

**Published:** 2024-06-27

**Authors:** Catalina Latorre Uriza, Nelly S. Roa, Juliana Velosa-Porras, Jean Carlos Villamil Poveda, Liliana Otero, Alvaro J. Ruiz, Francina María Escobar Arregoces

**Affiliations:** 1Centro de Investigaciones Odontológicas, Faculty of Dentistry, Pontificia Universidad Javeriana, Bogotá 110231, Colombia; juliana.velosa@javeriana.edu.co (J.V.-P.); jean.villamil@javeriana.edu.co (J.C.V.P.); lotero@javeriana.edu.co (L.O.); escobar.f@javeriana.edu.co (F.M.E.A.); 2Departamento de Medicina Interna, Faculty of Medicine, Pontificia Universidad Javeriana, Bogotá 110231, Colombia; aruiz@javeriana.edu.co; 3Departamento de Epidemiología Clínica y Bioestadística, Faculty of Medicine, Pontificia Universidad Javeriana, Bogotá 110231, Colombia

**Keywords:** periodontal disease, periodontitis, periodontal infection, periodontal debridement, inflammation mediators, cytokines, carotid intima-media thickness

## Abstract

A positive relationship has been reported between advanced periodontitis and carotid intima-media thickness (cIMT) measurement. The aim of this study was to investigate this relationship with parameters for periodontitis, such as PISA and systemic inflammation biomarkers. An observational descriptive cross-sectional study was conducted. A blood sample was collected from 75 subjects to analyze glucose, total cholesterol, HDL, LDL, and cytokine values. Increased cIMT was found in 32% of the patients with fewer teeth. Patients with periodontitis had a larger periodontal inflamed surface area (PISA) (*p* = 0.000) and had a 1.42-times-higher risk of having increased cIMT values compared to periodontally healthy individuals, though without a statistically significant association. Higher values in the left cIMT, IL-8, and TNF-α were found in men than in women with significant differences. In the multivariate analysis involving cytokines, age continues to be linked to increased cIMT values. INF-γ showed a trend towards a protective effect; as the IMT-M decreases, there is an increase in the expression of INF-γ, and a higher proportion of subjects with elevated INF-γ concentrations demonstrated normal IMT-C. This study did not find a statistically significant association between cIMT and periodontal disease, but the risk of having increased cIMT is 1.42-times higher for individuals with periodontitis.

## 1. Introduction

Periodontal disease is a condition of bacterial infectious etiology that triggers a chronic inflammatory response as a result of an unbalanced local immune response, which leads to the loss of connective tissue and bone support. Although periodontitis is an oral cavity disease, it has been associated with cardiovascular diseases [1,2], as demonstrated by recent epidemiological evidence showing an increased risk of vascular events' presentation and progression [3,4,5,6,7,8,9,10]. This association is mediated by bacteremia and increased plasma levels of systemic inflammation mediators linked to atherosclerosis [1,2,11].

Carotid intima-media thickness (IMT) is associated with aging and is considered a valid marker of subclinical atherosclerosis due to the cellular and molecular changes it induces. The measurement of this parameter has garnered considerable attention because it is strongly correlated with coronary artery disease and cerebral artery disease, serving as a good predictor for both vascular events [10,12]. Current evidence considers carotid IMT as a marker of arterial damage and a predictor of unfavorable outcomes, such as myocardial infarction [5,13,14].

Carotid IMT is primarily associated with traditional cardiovascular risk factors, such as age, sex, race, smoking, alcohol consumption, lack of exercise, high blood pressure, hypercholesterolemia, poor dietary patterns, and diabetes, among others [10]. However, these factors alone do not account for all the risks of coronary heart disease. It has been reported that over 60% of coronary heart disease cases are not explained by demographic and traditional cardiovascular risk factors. This is likely because of new risk factors, such as genetics, the presence of certain genotypes, immunological diseases, and inflammatory cytokines [15].

A positive relationship has been reported between advanced periodontitis and carotid IMT and between atherosclerotic vascular disease and periodontitis. Orlandi et al. in 2014 and Mustapha et al. in 2007 pointed out that periodontal disease, with its elevated inflammatory response triggered by periodontopathogenic microorganisms, could be significantly associated with an increase in carotid IMT [4,16]. Ding et al. in 2022, in a meta-analysis, reported that periodontitis, especially in an advanced state, was significantly associated with an increased risk of carotid IMT thickening. They also reported that periodontal treatment could help delay the progression of intima-media thickening [17]. 

Local periodontal infection triggers a systemic inflammatory response with an increase in leukocyte numbers [2], such as circulating monocytes; systemic cytokines like IL-1β, TNF-α, and IL-6, which have been proposed as potent inducers of C-reactive protein release in the liver [18]; LDL; HDL; and plasminogen activators, all of which play a role in the inflammatory process underlying atherosclerotic plaque formation [2,3,19,20,21].

For many years, it has been known that cytokines play a crucial role in the pathogenesis of periodontal disease by disrupting the balance of the inflammatory process. This occurs because there are more pro-inflammatory cytokines and fewer anti-inflammatory ones, inducing osteoclast activation that leads to the loss of bone support [22,23,24]. The cytokines most implicated in periodontal disease include TNF-α, IL-1β, IL-6, IL-17A, INF-γ, IL-8, and, to a lesser extent, IL-12 and IL-18. These have also been described as having a proatherogenic effect. In contrast, cytokines with antiatherogenic potential, such as IL-5, IL-10, IL-13, IL-19, IL-27, IL-33, IL-37, and TGF-β [25], are rarely mentioned in the context of periodontitis, except IL-33 [24]. Cytokines can alter various metabolic pathways, including lipid metabolism, potentially leading to an increase in total cholesterol levels and low-density lipoproteins (LDLs) [26,27,28]. Despite these findings, the specific role of each cytokine in periodontal disease and its association with an increased risk of carotid IMT thickening and cardiovascular disease remains unknown [2].

The World Health Organization (WHO) reveals that cardiovascular disease have been the leading cause of global mortality for the past 20 years. The number of deaths due to cardiovascular disease (CVD) has increased by over two million people since 2000, reaching nearly nine million people in 2019. Atherosclerotic cardiovascular disease (ASCVD) currently account for 16% of all deaths. More than half of the two million additional deaths have occurred in the WHO Western Pacific Region. In contrast, the European Region has experienced a relative decrease in CVD, with a 15% reduction in deaths [29].

On the other hand, the fourth National Oral Health Study in Colombia (ENSAB IV) emphasizes that the presence of periodontal disease contributes to exacerbating systemic problems, significantly affecting life expectancy. The majority of the Colombian population (61.8%) presents periodontitis in various degrees of severity, with moderate periodontitis being the most frequent at 43.46% [30].

Despite the extensive literature on the subject, the relationship between periodontal disease and IMT has not been completely elucidated [31], possibly due to genetic differences in the populations studied. While studies worldwide, including those involving Colombian residents, have evaluated the relationship between acute coronary syndrome and periodontal disease [32], few have explored the relationship between periodontal disease and carotid IMT in Colombian population.

Given the various methods available to analyze ASCVD, there has been no consensus in the literature about the best test for its diagnosis. The methods include systemic inflammatory biomarkers and non-invasive tests, like assessing endothelial dysfunction through brachial artery dilation and IMT [5]. Additionally, there are few reports using these tests on Colombian subjects. Therefore, the purpose of this study was to investigate the relationship between carotid IMT, clinical parameters (comorbidities and oral health), periodontal disease, and systemic inflammation biomarkers in a Colombian population.

## 2. Materials and Methods

An observational descriptive cross-sectional study was conducted. The study population consisted of adult patients over 30 years of age residing in Bogotá, Colombia, who attended consultations at the San Ignacio University Hospital and the Faculty of Dentistry at Pontifical Javeriana University (PUJ). Patients were sequentially recruited, and only those who had not received periodontal treatment or antibiotic treatment in the last three months were included. Diabetic patients and smokers were excluded. The project was approved by the research and ethics committee of the Faculty of Dentistry (CIEFOPUJ 007 of 2015).

After obtaining informed consent from all the patients, a blood sample was collected from each patient using a dry tube and an anticoagulant tube to obtain complete blood count values, glucose, total cholesterol, HDL, LDL, and systemic inflammation biomarkers through the evaluation of cytokines IL-1β, IL-2, IL-4, IL-6, IL-8, IL-10, IL-12p70, IL-17A, VEGF, INF-γ, and TNF-α. A portion of the serum was frozen in 1.5 mL vials at −20 °C until further use.

Serum cytokines were quantified using the Luminex system with the Milliplex Inflammation Human Cytokine kit Cat HCYTOMAG-60K (Merck KGaA, Darmstadt, Germany), as described above [33]. 

All patients underwent periodontal evaluations by completing the Periodontology annex of the PUJ Faculty of Dentistry. The periodontal examination (periodontal probing) was performed at six points around each tooth, for all teeth present in the mouth, with measurements recorded in millimeters. In this measurement, if bleeding occurred at each point, this was recorded and reported as “# Teeth with bleeding”. Measurements of up to 3 mm were considered normal sulci, while measurements of 4 mm and above were considered periodontal pockets. The presence of biofilm was measured by the O’Leary index. The periodontal diagnosis was made based on the Caton 2018 classification [34] (Table 1(A)). To carry out the statistical analysis as explained later, it is necessary to have a minimum number of individuals by diagnosis. For this reason, the study groups were categorized as follows: The healthy group included individuals with a diagnosis of healthy periodontium, reduced periodontium, and plaque-induced gingivitis in reduced periodontium. The periodontitis group included diagnoses of periodontitis stages III and IV, considered severe periodontitis. Periodontitis stages I and II were not included. Additionally, to assess periodontal inflammation, the PISA index (periodontal inflamed surface area in square millimeters (mm^2^)) was used.

Carotid ultrasonography was performed on the patients. The patient was in a supine position with a slight neck rotation to the contralateral side. The carotids were evaluated using an ultrasound machine with a 7.5 MHz transducer. Measurements were taken at the end of diastole, as systolic expansion in the lumen caused the intima-media thickness to become thinner. Measurements were taken at a length of 10 mm and the image quality was evaluated. IMT [5] was evaluated for both carotid arteries, with IMT-L representing the measurement of the left IMT and IMT-R representing the measurement of the right IMT. IMT-M corresponds to the average. 

Other associated risk factors were considered, including the percentage of biofilm, the number of teeth present in the mouth, and the number of teeth with bleeding. Additionally, the values of fasting glucose, total cholesterol, triglycerides, HDL, and LDL were included.

### Statistical Analysis

A description of the demographic characteristics and the results of periodontal evaluation of carotid IMT were presented, and systemic inflammation biomarkers were assessed using means, medians, standard deviations, and 95% confidence intervals. The normality of the inflammatory markers was assessed using the Shapiro–Wilk test, and to meet the test assumptions, the variables underwent a transformation via the Box–Cox method. The study’s power was calculated considering the difference in the proportion of patients with increased IMT values versus normal values, considering the periodontal diagnosis. It was found that the study has a power of 85%, with a minimum sample size required to detect this difference being 68 patients in total.

IMT was reported in absolute values, percentage change, and categorical terms (IMT-C) as normal or abnormal, using cut-off points defined in the CARMELA Study [35] for the city of Bogotá according to age and sex. 

For the analysis of outcomes, odds ratios were used. Adjustments were made for age and sex. Student’s *t*-tests or chi-squared tests were used for bivariate analysis, as appropriate. A difference was considered significant at *p* < 0.05 (two-tailed).

A logistic regression analysis was conducted to establish the association between systemic biomarkers, pocket depth levels, clinical attachment levels, and carotid IMT (normal or abnormal).

## 3. Results

A total of 75 patients were included, with an average age of 48 years (SD 1.46), of which 56% were male and 44% were female. Regarding oral conditions, the patients had an average of 24 teeth present in their mouths, and of these, an average of 17 exhibited bleeding. Additionally, 43.3% had dental plaque, 390.3 periodontal inflamed surface area (PISA), and 40% had some degree of periodontitis grouped as periodontitis. Increased IMT values were found in 32% of the patients (Table 1).

The healthy group included individuals with a diagnosis of healthy periodontium, reduced periodontium, and plaque-induced gingivitis in reduced periodontium. The periodontitis group included diagnoses of periodontitis stages III and IV, considered severe periodontitis; this was called the periodontitis group (Table 1(A)).

When analyzing by sex, it was found that men had higher values in the left carotid IMT compared to women, with statistically significant differences (0.74 vs. 0.58) (*p* = 0.039). HDL (53.5 vs. 42.2) and LDL (131.1 vs. 110.7) levels were higher in women, with statistically significant differences. Regarding oral conditions, no statistically significant differences were found, but it was observed that women had a lower percentage of dental plaque, a lower number of teeth with bleeding, and a lower prevalence of periodontitis than men. Additionally, concerning the presence of increased IMT, the number of women with increased IMT was lower than that found in men (Table 2).

Regarding the groups, based on whether they had periodontitis or not, differences were found in age, with the healthy group being younger compared to the group with periodontitis (11 years difference). In terms of the number of teeth present in the mouth, the healthy group had an average of 25 teeth, while those with periodontitis had an average of 21 teeth (*p* = 0.001). The percentage of dental plaque was twice as high in the group with periodontitis (62.6% vs. 30.7%), and the number of teeth with bleeding was higher in this group (periodontitis) as well as the PISA value, which was 3.47-times greater than the healthy group (*p* = 0.000) (Table 3).

When analyzing the results based on having normal or increased values of IMT, according to the values established in the CARMELA study [35], significant differences were found in terms of age, with higher values for patients of an older age on average for right IMT, left IMT, and overall IMT. Regarding the number of teeth present, the group with increased IMT values had a lower number of teeth compared to the group with normal IMT values (21.5 vs. 24.6) (Table 4). 

Regarding the cytokine analysis by sex, significant differences were found in IL-8 and TNF-α, with higher values in men than in women (*p* = 0.04 and *p* = 0.010, respectively) (Table 5). However, no statistically significant differences were found in the presence of the analyzed serum cytokines concerning periodontal condition between the healthy and periodontitis groups (Table 6), nor when comparing normal or increased intima thickness (Table 7). 

In addition, when cytokines were evaluated according to the cut-off point and IMT (normal and increased), no association was found between cytokines and increased intima-media thickness (Table 8).

In the multivariate analysis, in Model 1, it was found that patients with periodontitis had a 1.42-times-higher risk of having increased IMT values compared to periodontally healthy individuals, but this association was not statistically significant (*p* = 0.482). In Model 2, when controlling for sex and age, the risk increased to 2.66 (95% CI 0.25; 28.3), but it remained statistically insignificant (*p* = 0.398). Model 3 shows that the only variable associated with increased IMT values is age (OR: 1.06, 95% CI 1.00; 1.13). In Model 4, when including cytokines, age remains associated with increased IMT values (OR: 1.12, 95% CI 1.03; 1.21), and INF-γ shows a trend towards a protective effect against increased IMT values (*p* = 0.040; OR: 0.76, 95% CI 0.59; 0.98) (Table 9). As can be seen in Figure 1, as the IMT-M is lower in the same subjects, the expression of INF-γ is increased (Figure 1A) and there are more subjects with high INF-γ concentrations who have normal IMT-C values (Figure 1B).

## 4. Discussion

The present research demonstrated that, in a group of adults who visited the Faculty of Dentistry, men had higher values in the left IMT compared to women, with a statistically significant difference (0.74 vs. 0.58) (*p* = 0.039). These findings support those of Yu et al. (2014), where out of 245 patients with abnormal IMT, 150 were men and 95 were women [36]. In the same vein, Ta-Chen Su et al., in 2012, when assessing risk factors and sex differences in common carotid artery IMT in a Chinese population, reported that IMT was higher in men than in women [37].

Analyzing the relationship between IMT and age concerning having normal or elevated IMT values according to the thresholds defined in the CARMELA study [35], the present study found that, in patients of a older age, the average values of right IMT, left IMT, and global IMT were increased with statistically significant differences. These findings are consistent with Van den Munckhof et al. (2018), who, in a systematic review, found a strong linear relationship between age and IMT in healthy and asymptomatic individuals, suggesting that IMT linearly progresses with age. They also noted that individuals with cardiovascular risk or CVD showed higher IMT values at all age groups, but the annual increase in IMT was comparable to that of healthy and asymptomatic individuals. This suggests that CVD and cardiovascular risk affect IMT, but not the linear relationship between age and IMT. This systematic review highlighted the presence of a constant linear increase in IMT throughout life, and the presence of CVD or cardiovascular risk does not affect the direction and linear nature of this relationship [38].

Regarding the number of present teeth, the group with elevated IMT had fewer teeth compared to the group with normal IMT (21.5 vs. 24.6). Similar results have been reported by Almoosawy et al. (2021). They conducted a systematic review on the oral health status of people diagnosed with peripheral vascular disorders (PVDs) and found that participants with PVD had compromised oral health in various measures, with worse periodontal health, a higher number of missing teeth, and a higher prevalence of edentulism compared to participants without PVD [39]. Likewise, Jung et al. (2014) conducted a study on the relationship between periodontal disease and subclinical atherosclerosis in 5404 individuals aged 50 years and older and found that the number of missing teeth was associated with an increased IMT [40].

Additionally, Fukuhara et al. (2023) reported the associations between tooth loss, periodontitis, and carotid IMT in a cross-sectional study that included 9778 participants from the Nagahama study. In a multivariable analysis adjusted for conventional risk factors, they identified a significant determinant of intima-media thickness to be many missing teeth (fewer than nine remaining teeth), which was related to prolonged inflammation indicative of the highest stage of periodontitis [41].

When analyzing the average IMT values in relation to periodontal disease, the present study observed that the trend of the average arterial intima-media value was higher for patients with periodontitis (healthy 0.60, periodontitis 0.67 mm), although without statistically significant differences. In a similar way, Tonetti et al. (2009) reported that patients with severe generalized chronic periodontitis had a higher risk of increased carotid IMT than those with less severe and less widespread disease, suggesting a dose-dependent effect of periodontal disease on arterial intima-media thickness, and this effect was not statistically significant [42]. Furthermore, Orlandi et al. (2014), in a systematic review and meta-analysis, evaluated the association between carotid IMT and periodontal disease, reporting that patients with periodontal disease had a higher average carotid IMT (0.08 mm) than those without periodontal disease [4]. While the magnitude of the difference might seem small, it is relevant to note that the range of progression in carotid IMT in the general population is from 0.001 to 0.030 mm per year. The evidence suggests that, for every 0.1 mm difference in carotid IMT, the relative risk of acute myocardial infarction increases by 1.15 times [4]. More recently, Lamprecht et al. (2022), in a cross-sectional analysis of the association of periodontitis with carotid IMT and atherosclerotic plaques, reported that the median of IMT ≥ 1 mm and the prevalence were significantly higher in men and women with chronic periodontitis (*p* < 0.001). Moreover, severe chronic periodontitis was associated with increased IMT and a higher prevalence of carotid plaques, independent of common risk factors [43].

The analysis of the systemic inflammatory response, through the measurement of inflammatory cytokines in relation to periodontal condition (healthy and periodontitis) and IMT (normal and increased), found no statistically significant differences. This was also the case when cytokines were evaluated according to the cut-off point reported in the literature [44,45,46,47,48,49,50]. No association was found between cytokines and increased intima-media thickness (normal and increased). However, higher values of IL-8 and TNF-α were found in male subjects (*p* = 0.04 and *p* = 0.010, respectively) with increased values of left IMT (*p* = 0.039), cytokines that have been described to have a proatherogenic effect [25]. These findings propose these cytokines as systemic biomarkers of susceptibility to increased IMT values.

Recently, the relationship between periodontal disease and CVD through the increase in systemic inflammatory response that periodontitis induces has been reported. Delange et al. (2018), in their study on periodontal disease and its connection with systemic CVD markers in young American Indian/Alaska Native individuals, compared interleukin-6 (IL-6) and C-reactive protein (CRP) levels across the severity of periodontal disease status among younger adults aged 21 to 43 years. The results showed that severe periodontitis was significantly associated with elevated IL-6 levels compared to those without periodontitis or with mild periodontitis before controlling for other variables, but it lacked significance after controlling for sex, body mass index, smoking, and high-density lipoprotein. Moderate periodontal disease was positively associated with IL-6 levels after controlling for potential confounding factors [18].

Periodontitis might lead to transient vascular inflammation and endothelial dysfunction. Furthermore, the periodontitis-induced effects on endothelial function could be a potential early step toward the development of CVD [51,52]. Periodontitis is thought to cause low-grade systemic inflammation due to the production and release of inflammatory markers, including TNF-α and IL-6, into the bloodstream, which negatively affect endothelial function. Endothelial dysfunction is considered a precursor to atherosclerosis and CVD [52]. In our study, it was interesting to find that, in the multivariate analysis (Model 4), IFN-γ appeared to show a trend toward a protective effect against increased IMT. While IFN-γ has been typically categorized as proinflammatory and proatherogenic [25], this finding might suggest that elevated levels of IFN-γ can indicate a lower risk of atherosclerosis, such as an innate immune response. However, more research would be needed to confirm this and to understand the underlying mechanisms.

The discrepancies in the analysis of the systemic inflammatory response through cytokines in relation to periodontal condition (healthy and periodontitis) and IMT (normal and increased) across various studies are likely due to differences in the populations studied, including factors such as race, age groups, and geographical locations [31]. On the other hand, a cut-off point for each cytokine as well as its detection level may be different, since they can be detected by various techniques, such as Luminex, ELISA and Flow Cytometry [44,45,46,47,48,49,50]. 

To gain a more precise understanding of the role of cytokines in the relationship between carotid IMT and periodontal disease, future research is necessary to investigate the correlations between clinical parameters of periodontal disease and the balance between pro-inflammatory cytokines, such as IFN-γ, IL-17A, IL-1β, IL-2, IL-6, and IL-8; anti-inflammatory IL-10 and IL-4; and immunoregulatory cytokines, such as IL-27, IL-23, IL-22, IL-35, TGF-β, IL-13, VEGF, IL-5, and IL-29, involved in cardiovascular disease. These can be evaluated in gingival crevicular fluid and systemic circulation and to establish biochemical and hematological parameters associated with inflammation and carotid IMT as indicators of atherosclerosis [25]. A biological model could be constructed with background information from the literature and the results obtained in this study as starting a point (Figure 2).

The comparison between periodontally healthy and periodontitis groups in our research showed that patients with periodontitis had a 1.42-times-higher risk of having increased IMT than periodontally healthy individuals, but this association was not statistically significant (*p* = 0.481). When controlling for sex and age, the risk increased to 2.66 (95% CI 0.25 to 28.3), also without statistical significance (*p* = 0.398). Similar results were reported by Ding et al. (2022) in their meta-analysis [14]. Additionally, Nitya et al. (2020) concluded that periodontal disease and poor oral hygiene were more severe in subjects with IMT > 1 mm [53].

Upon conducting the analysis of PISA (Periodontal Inflamed Surface Area), the current study found that patients with periodontitis had a larger inflamed surface area (681.7 mm²) with a statistically significant difference (*p* = 0.000). Leira et al., in 2017, indicated that a PISA value ≥ 130.33 mm^2^ could identify patients with periodontitis [54]. The Periodontal Inflamed Surface Area (PISA) has been associated with systemic inflammation. In this context, Leira et al., in 2019, studied the association between periodontitis and higher circulating levels of systemic inflammation and endothelial dysfunction biomarkers in patients with lacunar infarcts. They reported that, for patients with worse outcomes, PISA correlated positively with IL-6 (r = 0.738, *p* < 0.001), PTX3 (r = 0.468, *p* = 0.008), sTWEAK (r = 0.771, *p* < 0.001), and Aβ 1–40 (r = 0.745, *p* < 0.001), suggesting a link between periodontitis, inflamed periodontal area, systemic inflammatory response, and alteration in vascular endothelial function in patients with lacunar infarcts [55]. Similarly, Onabanjo et al., in 2023, analyzed the association between the Periodontal Inflamed Surface Area (PISA) and systemic inflammatory biomarkers in patients with chronic kidney disease in pre-dialysis. They pointed out a possible association between periodontal inflammation and an increase in hsCRP as a marker of systemic inflammatory load [56].

Our study has several limitations that should be considered. First, the results of convenience sampling we used must be confirmed in further studies. Second, as an observational study, it cannot support a causal relationship between carotid IMT and periodontal disease, but can suggest potential inflammation biomarkers present in both conditions. Although the results of this study have a power of 85%, with a minimum sample size of 68 patients required to detect this difference, to improve the sample size and include all stages and degrees of periodontitis, systemic and local biomarkers in gingival crevicular fluid involved in this association could be shown. Third, we did not control the statistical analysis for anti-hypertensives medications and other drugs that might have affected both periodontal and ultrasonographic examinations. This would help explain why the levels of cytokines with respect to the cut-off point were increased in some subjects, but not high enough to reflect an evident systemic inflammatory state. Therefore, finding these levels of cytokines almost at normality and without differences between periodontally healthy subjects suggests that the treatment for the primary disease of carotid intima-media thickness can control the systemic inflammatory process and mask this association. Additionally, and hypothetically for future studies, this primary treatment could affect the healing process and prognosis of periodontitis if changes in biological markers in gingival crevicular fluid (GCF) pathognomonic of periodontal disease are observed.

Fourth, the present study focused on cytokines as biomarkers of inflammation, but for logistical and budgetary reasons, other already identified biomarkers were not taken into account, such as monocyte chemoattractant protein-1 (MCP-1), soluble CD40 ligand, serum amyloid A (SAA), selectins (E-selectin and P-selectin), myeloperoxidase (MPO), matrix metalloproteinases (MMPs), cellular adhesion molecules, intercellular adhesion molecule 1 (ICAM-1), vascular adhesion molecule 1 (VCAM-1), placental growth factor (PlGF), and A2 phospholipases, as well as other important known risk factors. However, this does not diminish the findings regarding the investigated cytokines.

There are other important cardiovascular risk factors that have also been reported in the literature, such as C-reactive protein (CRP), serum uric acid levels (SUAs), and estimated glomerular filtration rate (eGFR). C-reactive protein (CRP) is considered a cardiovascular risk factor since its elevated levels have been associated with adverse cardiovascular outcomes, like acute coronary syndrome (ACS), as well as in the initiation and development of atherosclerotic plaque [57,58]. Inflammatory lesions, such as those caused by periodontitis, can also contribute to an increase in systemic inflammation and elevated CRP levels [1]. In this regard, D’Aiuto et al. reported that intensive periodontal therapy could lead to a significant reduction in lymphocyte counts, CRP levels, IL-6, and LDL cholesterol [59]. This research did not analyze CRP, but it would be worthwhile to include it in future studies to assess cardiovascular risk from this marker of systemic inflammation.

Another important predictor of cardiovascular (CV) morbidity and mortality is serum uric acid (UA) [60]. Canterin et al. found that septic UA is an explanatory factor for the thickening of the intima-media and the Beta stiffness index [61]. It is noteworthy that more recent research links elevated levels of uric acid with periodontitis [62]. Moore et al. reported that increased salivary concentrations of uric acid suggest oxidative stress and the progression of periodontal disease. Uppin and Varghese conducted a systematic review and meta-analysis to analyze the concentration of UA in the serum, saliva, and gingival crevicular fluid (GCF) of subjects with and without periodontal disease. They reported an increase in serum UA levels in periodontal disease, but salivary UA levels decreased in these patients. They noted that it was not clear why UA levels were the opposite in the blood and saliva of patients with periodontal disease, suggesting the need for more research in this area [63].

The relationship of serum UA levels both with the risk of increase in arterial intima-media thickness and the increase in serum levels in patients with periodontitis should motivate future studies to analyze the relationship between periodontal disease, arterial IMT, and serum uric acid levels to further explore if SUA could be used as a diagnostic biomarker to assess the risk or progression of periodontitis and atherosclerotic cardiovascular disease.

The relationship of periodontal disease with cardiovascular disease and kidney disease should also be studied, as renal patients show an increase in oxidative stress, chronic inflammation, and the release of extracellular vesicles (EVs), which cause endothelial damage and the development of various cardiovascular diseases (CVDs) [64]. In this regard, Chambrone et al. conducted a systematic review of the literature and evaluated the association between periodontitis and chronic kidney disease (CKD), as well as the effect of periodontal treatment (PT) on estimated glomerular filtration rate (eGFR). They reported an association between periodontitis and CKD. Similarly, this result was supported by pooled estimates (OR: 1.65, 95% confidence interval: 1.35, 2.01, *p* < 0.00001, χ(2) = 1.70, I(2) = 0%). They reported positive results in the estimated glomerular filtration rate (eGFR) after periodontal treatment [65].

Given that this research did not manage to analyze all the factors related to cardiovascular risk, future studies examining these aspects must be conducted.

It is important to acknowledge that, while these studies reveal associations and trends, the relationships between periodontal disease, cardiovascular health, and inflammatory markers are complex and multifaceted. Further research is needed to elucidate the specific mechanisms and causal relationships underlying these observations. On the other hand, individual factors and potential confounding variables should be considered when interpreting the results of epidemiological studies.

## 5. Conclusions

In summary, while this study did not find a statistically significant association between IMT and periodontal disease in the Colombian population studied, the results suggest several important observations.

The risk of having increased IMT (a marker of atherosclerosis) is 1.42-times higher for individuals with periodontitis compared to those without periodontitis. This finding is supported by the observation that individuals with increased IMT tend to have less teeth, which is a known indicator of periodontal health.

Men had higher values of left IMT compared to women, and as age increased, the average IMT values tended to be higher.

The study found no significant differences in the levels of systemic inflammation biomarkers (cytokines) with respect to periodontal condition (healthy and periodontitis), although men had higher values of IL-8 and TNF-α, which may be related to the higher left IMT values observed in men, proposing these cytokines as systemic biomarkers of susceptibility to increased IMT values. 

For this population, serum INF-γ showed a trend towards a protective effect against increased IMT, suggesting that it might play a role in the pathophysiology of atherosclerosis.

## Figures and Tables

**Figure 1 biomedicines-12-01425-f001:**
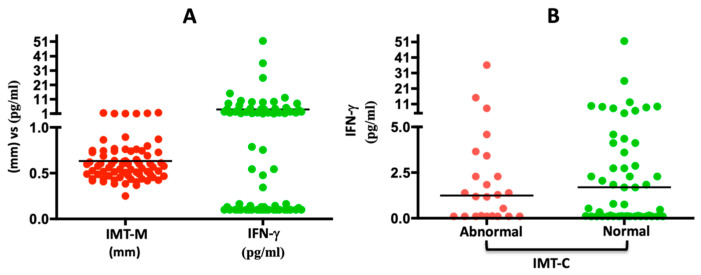
INF-γ shows a trend towards a protective effect against increased IMT values. (**A**) Each red dot represents the average in millimeters (mm) of IMT-M obtained by measuring IMT-R with IMT-L for every subject included in the study, while each green dot represents their respective INF-γ concentration. The horizontal line illustrates the mean value within each IMT-M or INF-γ group. (**B**) Each dot represents the expression of IFN-γ in subjects with abnormal (red) or normal (green) IMT-C values.

**Figure 2 biomedicines-12-01425-f002:**
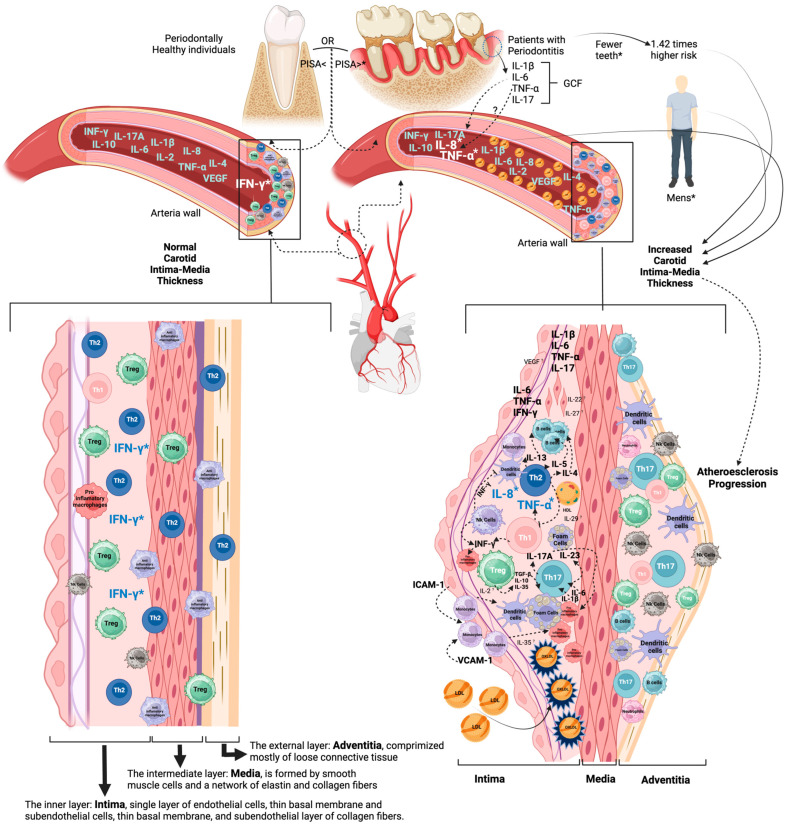
Relationship between carotid intima-media thickness, periodontal disease, and systemic inflammation biomarkers, biological model. Atherosclerosis is characterized by the formation of atherosclerotic plaques in the subendothelial layer, smooth muscle cell proliferation, accumulation of activated immune cells, and thickening of adventitia alongside of plaque formation; carotid intima-media thickness is a predictive marker of atherosclerosis. Among the cytokines produced during periodontal disease, such as IL-1β, IL-6, IL-8, IL12, IL-17A, IL-18, TNF-α, and INF-γ [22,23,24], some have been described to have a proatherogenic effect. In contrast, IL-5, IL-10, IL-13, IL-19, IL-27, IL-33, IL-37, and TGF-β may have antiatherogenic potential [25]; in periodontitis, only IL-33 [24] has been found to play a possible role. The increase in total cholesterol levels and Low-Density lipoproteins (LDLs) can be explained by the deregulated participation of these cytokines in lipid metabolism [26,27,28]. The specific role of individual cytokines in this relationship has been described by several authors [2,3,18,19,20,21] and in this work (*). An explanatory biological model is presented. GCF: gingival crevicular fluid. Figure adapted from [25] and Created with BioRender.com.

**Table 1 biomedicines-12-01425-t001:** Characteristics of the study population (*n* = 75).

**Variable**	**Mean (SD)**		**95% CI**
Age	48.4 (1.46)		45.4–51.3
IMT-R	0.61 (0.02)		0.56–0.66
IMT-L	0.65(0.03)		0.58–0.72
IMT-M	0.63 (0.02)		0.57–0.69
Baseline glycemia	97.0 (3.95)		89.1–104.9
Total cholesterol	201.4 (4.86)		191.7–211.1
Triglycerides	158.6 (13.2)		132.3–184.9
cHDL	48.5 (1.69)		45.1–51.9
cLDL	122.1 (4.41)		113.3–130.9
# Teeth	23.6 (0.64)		22.3–24.9
Biofilm	43.4 (3.22)		37.0–49.9
# Teeth with bleeding	17.1 (0.91)		15.2–18.9
PISA	390.3 (50.3)		289.9–490.7
INF-γ	3.76 (0.92)		1.93–5.60
IL-10	1.69 (0.83)		0.03–3.36
IL-17A	3.25 (0.59)		2.07–4.43
IL-1β	0.69 (0.17)		0.34–1.04
IL-2	0.32 (0.12)		0.07–0.57
IL-6	8.24 (3.76)		0.74–15.7
IL-8	11.9 (2.02)		7.95–16.0
TNF-α	11.2 (0.70)		9.81–12.6
IL-12p70	1.72 (0.43)		0.85–2.59
IL-4	175.1 (64.2)		47.1–303.1
VEGF	46.1 (4.67)		36.8–55.4
Sex	%		
Female	56		44.3–66.9
Male	44		33.0–55.6
Group ^Ψ^	%		
Healthy	60		48.3–70.6
Periodontitis	40		29.3–51.6
IMT	%		
Normal	68		56.3–77.7
Increased	32		22.2–43.6
**^Ψ^ (A). Classification of study groups according to periodontal diagnosis (Workshop 2017)**			
	** *n* **	**%**	**Study Groups**
Reduced healthy periodontium	2	2.67	Healthy group
Biofilm-induced gingivitis on reduced periodontium in a periodontally treated patient	43	57.33	
Periodontitis stage III	16	21.33	Periodontitis group
Periodontitis stage IV	14	18.67	
TOTAL	75	100	

IMT: Carotid Intima-Media Thickness; IMT-L: Left Carotid Intima-Media Thickness; IMT-R: Right Carotid Intima-Media Thickness; IMT-M: corresponds to the average of IMT-L and IMT-R; LDLs: Low-Density Lipoproteins; HDLs: High-Density Lipoproteins; # Teeth: number of teeth present; # Teeth with bleeding: number of bleeding teeth; PISA: periodontal inflamed surface area. Study groups were grouped as described in (A) ^Ψ^.

**Table 2 biomedicines-12-01425-t002:** Bivariate analysis by sex.

Variable	Female	Male	*p*-Value
Age	49.1 (1.90)	47.4 (2.28)	0.569
IMT-R	0.57 (0.02)	0.66 (0.05)	0.105
IMT-L	0.58 (0.02)	0.74 (0.06)	0.039 *
IMT-M	0.57 (0.02)	0.70 (0.05)	0.053
Baseline glycemia	95.0 (5.97)	99.6 (4.85)	0.566
Total cholesterol	208.3 (6.13)	192.7 (7.66)	0.111
Triglycerides	126.9 (10.5)	198.9 (25.4)	0.012
cHDL	53.5 (2.35)	42.2 (1.97)	0.000 *
cLDL	131.1 (5.09)	110.7 (7.27)	0.021 *
# Teeth	23.6 (0.80)	23.6 (1.04)	0.977
Biofilm	38.9 (3.82)	49.3 (5.37)	0.110
# Teeth with bleeding	16.1 (1.18)	18.3 (1.42)	0.225
PISA	354.0 (56.2)	436.5 (89.8)	0.420
Group	#	#	0.237
Healthy	28	17	
Periodontitis	14	16	
IMT-C	#	#	0.319
Normal	31	20	
Increased	11	13	

IMT: Carotid Intima-Media Thickness; IMT-L: Left Carotid Intima-Media Thickness; IMT-R: Right Carotid Intima-Media Thickness; IMT-M: corresponds to the average of IMT-L and IMT-R; LDLs: Low-Density Lipoproteins; HDLs: High-Density Lipoproteins; # Teeth: number of teeth present; # Teeth with bleeding: number of bleeding teeth; PISA: periodontal inflamed surface area. * *p* < 0.05.

**Table 3 biomedicines-12-01425-t003:** Bivariate analysis by periodontal condition.

Variable	Healthy	Periodontitis	*p*-Value
Age	43.9	55.1	0.000 *
IMT-R	0.60	0.63	0.554
IMT-L	0.61	0.70	0.168
IMT-M	0.60	0.67	0.275
Baseline glycemia	92.4	103.9	0.157
Total cholesterol	198.3	206.1	0.432
Triglycerides	155.2	163.6	0.764
cHDL	47.2	50.5	0.331
cLDL	120.3	124.8	0.619
# Teeth	25.2	21.1	0.001 *
Biofilm	30.7	62.6	0.000 *
# Teeth with bleeding	15.3	19.7	0.019 *
PISA	196.0	681.7	0.000 *
Sex	#	#	0.237
Female	28	14	
Male	17	16	
IMT-C	#	#	0.614
Normal	32	19	
Increased	13	11	

IMT: Carotid Intima-Media Thickness; IMT-L: Left Carotid Intima-Media Thickness; IMT-R: Right Carotid Intima-Media Thickness; IMT-M: corresponds to the average of IMT-L and IMT-R; LDLs: Low-Density Lipoproteins; HDLs: High-Density Lipoproteins; # Teeth: number of teeth present; # Teeth with bleeding: number of bleeding teeth; PISA: periodontal inflamed surface area. * *p* < 0.05.

**Table 4 biomedicines-12-01425-t004:** Bivariate analysis by IMT-C.

Variable	Normal	Increased	*p*-Value
AGE	45.6	54.3	0.004 *
IMT-R	0.50	0.84	0.000 *
IMT-L	0.52	0.92	0.000 *
IMT-M	0.51	0.88	0.000 *
Baseline glycemia	98.4	93.9	0.592
Total cholesterol	201.0	202.1	0.916
Triglycerides	140.8	196.4	0.082
cHDL	51.1	43.0	0.024
cLDL	123.2	119.9	0.729
# Teeth	24.6	21.5	0.025 *
Biofilm	41.4	47.8	0.358
# Teeth with bleeding	17.9	15.2	0.165
PISA	363.7	446.8	0.425

IMT: Carotid Intima-Media Thickness; IMT-L: Left Carotid Intima-Media Thickness; IMT-R: Right Carotid Intima-Media Thickness; IMT-M: corresponds to the average of IMT-L and IMT-R; LDLs: Low-Density Lipoproteins; HDLs: High-Density Lipoproteins; # Teeth: number of teeth present; # Teeth with bleeding: number of bleeding teeth; PISA: periodontal inflamed surface area. * *p* < 0.05.

**Table 5 biomedicines-12-01425-t005:** Cytokines by sex.

Variable	Female Median	Male Median	*p*-Value
IFN-γ	0.411	2.06	0.052
IL-10	0.08	0.08	0.427
IL-17A	1.54	2.38	0.166
IL-1β	0.04	0.04	0.760
IL-2	0.04	0.04	0.636
IL-6	0.02	0.02	0.817
IL-8	4.85	9.71	0.004 *
TNF-α	9.01	11.9	0.010 *
IL-12p70	0.25	0.53	0.359
IL-4	4.4	4.4	0.700
VEGF	32.0	36.3	0.394

* Mann–Whitney U test.

**Table 6 biomedicines-12-01425-t006:** Cytokines and periodontal condition.

Variable	Healthy Median	Periodontitis Median	*p*-Value
IFN-γ	1.18	1.34	0.673
IL-10	0.08	0.08	0.291
IL-17A	2.14	1.76	0.319
IL-1β	0.04	0.04	0.561
IL-2	0.04	0.04	0.652
IL-6	0.02	0.02	0.294
IL-8	5.98	8.79	0.770
TNF-α	9.27	10.6	0.230
IL-12p70	0.28	0.28	0.879
IL-4	4.4	4.4	0.662
VEGF	36.1	31.1	0.569

Mann–Whitney U test.

**Table 7 biomedicines-12-01425-t007:** Cytokines by normal or increased IMT-C.

Variable	Normal IMT	Increased IMT	*p*-Value
IFN-γ	1.68	1.24	0.691
IL-10	0.08	0.08	0.546
IL-17A	2.38	1.92	0.433
IL-1β	0.04	0.04	0.863
IL-2	0.04	0.04	0.895
IL-6	0.02	0.02	0.964
IL-8	5.98	7.00	0.535
TNF-α	8.97	11.3	0.063
IL-12p70	0.28	0.41	0.492
IL-4	4.4	4.4	0.994
VEGF	36.1	31.7	0.846

Mann–Whitney U test.

**Table 8 biomedicines-12-01425-t008:** Cytokine analysis according to cut-off points.

	Cut-Off Point pg/mL	Normal IMT	Increased IMT	*p* Value
IFN-γ	≤0.84	25	10	0.780
	>0.84	26	12	
IL-10	≤2.130	47	22	0.942
	>2.130	4	2	
IL-17A	≤5.75	47	23	0.551
	>5.75	4	1	
IL-1β	≤5	49	23	0.959
	>5	2	1	
IL-2	≤1.668	48	22	0.245
	>1.668	3	0	
IL-6	≤0.744	41	20	0.760
	>0.744	10	4	
IL-8	≤12.35	39	18	0.889
	>12.35	12	6	
TNF-α	≤1.323	1	0	0.489
	>1.323	50	24	
IL-12 P70	<1.0	38	14	0.1564
	>1.0	13	10	
IL-4	≤1.765	0	0	0.602
	>1.765	51	24	
VEGF	≤30	23	12	0.639
	>30	29	12	

Chi-squared test.

**Table 9 biomedicines-12-01425-t009:** Logistic regression—periodontal disease.

Variable	OR	95% CI	*p*-Value
MODEL 1			
Periodontitis	1.42	0.52; 3.84	0.482
MODEL 2			
Periodontitis controlling for age and sex	2.66	0.25; 28.43	0.398
MODEL 3			
Periodontitis	0.61	0.09; 3.92	0.606
Male	1.77	0.43; 7.30	0.424
Age	1.06	1.00; 1.13	0.022 *
Baseline glycemia	0.97	0.93; 1.01	0.150
Total cholesterol	1.83	0.49; 6.85	0.368
Triglycerides	0.88	0.68; 1.15	0.386
cHDL	0.53	0.14; 1.98	0.348
cLDL	0.54	0.14; 2.05	0.371
Biofilm	1.00	0.98; 1.03	0.525
# Teeth with bleeding	0.93	0.85; 1.01	0.126
PISA	1.00	0.99; 1.00	0.536
MODEL 4			
Periodontitis	0.81	0.07; 8.76	0.863
Male	2.73	0.41; 17.8	0.294
Age	1.12	1.03; 1.21	0.004 *
Baseline glycemia	0.95	0.90; 1.01	0.141
Total cholesterol	3.91	0.70; 21.6	0.119
Triglycerides	0.76	0.54; 1.07	0.129
cHDL	0.24	0.04; 1.36	0.109
cLDL	0.25	0.04; 1.43	0.122
Biofilm	1.02	0.98; 1.06	0.287
# Teeth with bleeding	0.89	0.80; 1.00	0.053
PISA	1.00	0.99; 1.00	0.597
IFN-γ	0.76	0.59; 0.98	0.040 *
IL-10	0.93	0.76; 1.13	0.498
IL-17a	1.34	0.97; 1.85	0.072
IL-1β	0.63	0.05; 7.25	0.711
IL-2	0.31	0.00; 15.6	0.564
IL-6	1.06	0.96; 1.16	0.193
IL-8	0.92	0.82; 1.04	0.228
TNF-α	1.02	0.85; 1.23	0.800
IL-12p70	1.54	0.85; 2.79	0.148
IL-4	0.99	0.99; 1.00	0.586
VEGF	0.99	0.96; 1.01	0.448

* Logistic regression. Model 1: Group (healthy and periodontitis); Model 2: Group (healthy and periodontitis) controlling for age and sex; Model 3: Group (healthy and periodontitis), age, sex, glucose, total cholesterol, triglycerides, HDL, LDL, dental plaque, # Teeth with bleeding: number of bleeding teeth, and PISA; Model 4: Group (healthy and periodontitis), age, sex, glucose, total cholesterol, triglycerides, HDL, LDL, dental plaque, number of bleeding teeth, PISA, and cytokines. # Teeth with bleeding: number of bleeding teeth.

## Data Availability

The data of this research are available in the archives of the Pontificia Universidad Javeriana.

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
