# Peer review of "Relationship between Carotid Intima-Media Thickness, Periodontal Disease, and Systemic Inflammation Biomarkers in an Adult Population"

_biomedicines, 2024, doi:10.3390/biomedicines12071425_

Round 1

Reviewer 1 Report (Previous Reviewer 1)

Comments and Suggestions for Authors

Now it can be accepted

Author Response

We appreciate the positive review reports.

Thank you.

Reviewer 2 Report (Previous Reviewer 2)

Comments and Suggestions for Authors

The authors have improved their paper. However I've yet some minor comments:

- The acronym "PISA" should be spelled in the abstract

- The authors have to more deeply discuss the limitation of their study (this was already asked in the first review of this paper but yet avoided). In particular, first line inflammatory parameters such as VES and CRP have not been considered, as well as a number of first level CV risk factors (for instance, eGFR, SUA). 

Author Response

Revisor 2 comments:

- The acronym "PISA" should be spelled in the abstract

Answer: the meaning of PISA was included into abstract, on line 17 and 18.

- The authors have to more deeply discuss the limitation of their study (this was already asked in the first review of this paper but yet avoided). In particular, first line inflammatory parameters such as VES and CRP have not been considered, as well as a number of first level CV risk factors (for instance, eGFR, SUA). 

Answer: First line inflammatory parameters such as VES, CRP, as well as a number of first level CV risk factors were included into our limitations of this study into discussion, on line 427 – 475.

Thank you for your comments, improve our study.

Reviewer 3 Report (Previous Reviewer 3)

Comments and Suggestions for Authors

I thank you for your polite responses and revisions.

I checked them and have agreed with your revisions.

I think this manuscript can be acceptable for this journal by these revisions.

Author Response

We appreciate the positive review reports.

Thank you.

This manuscript is a resubmission of an earlier submission. The following is a list of the peer review reports and author responses from that submission.

Round 1

Reviewer 1 Report

Comments and Suggestions for Authors
  • Describe how bleeding and the presence of biofilm were measured. 

  • How was established the sample size? 

  • Table 4 shows bivariate analysis by IMT-C but should include oral variables.

Reviewer 2 Report

Comments and Suggestions for Authors

I’ve read with attention the paper of Latorre Uriza et al. 

- Abstract: The first sentence has to be rewritten.

- Methods: the sample size is very small, too small to drive any conclusion. First line inflammatory parameters such as VES and CRP have not been considered, as well as a number of first level CV risk factors (for instance, eGFR, SUA). Moreover, the comparison statistics has been managed as all the variables were normally distributed.

- Discussion: it is very long and unfocused, while the authors avoided to discuss the main limitation of the study (see above).

Comments on the Quality of English Language

The quality of English language is overall good. Minor erros or typos can be easily corrected.

Reviewer 3 Report

Comments and Suggestions for Authors

The authors suggest several important observations while this study did not find a statistically significant association between IMT and periodontal disease in the Colombian population studied.

These observations are important and useful for the regulation of periodontitis, but a more significant relation should be shown for such a clinical report.

So, I suggest that the authors add some data and perform more detailed analyses to report more informative results as follows.

1. Suggestion: I think the parameters for periodontitis are few. Some reports use PISA [Periodontal inflamed surface area; Nesse W et al. (J Clin Periodontol 2008)] to investigate the relationship between periodontitis and systemic disease. Please add PISA and other parameters, for example, serum antibody titer or bacterial counts for periodontal pathogenic bacteria.

2. Suggestion: I recommend the authors analyze by severity of periodontitis.

3. Question: Does the treatment for primary disease of carotid intima-Media thickness affect the healing process and prognosis of periodontitis?